# Nutrigenomics and Nutrigenetics in Metabolic- (Dysfunction) Associated Fatty Liver Disease: Novel Insights and Future Perspectives

**DOI:** 10.3390/nu13051679

**Published:** 2021-05-15

**Authors:** Marcello Dallio, Mario Romeo, Antonietta Gerarda Gravina, Mario Masarone, Tiziana Larussa, Ludovico Abenavoli, Marcello Persico, Carmelina Loguercio, Alessandro Federico

**Affiliations:** 1Department of Precision Medicine, University of Campania Luigi Vanvitelli, Via S. Pansini 5, 80131 Naples, Italy; mario.romeo@unicampania.it (M.R.); antoniettagerarda.gravina@unicampania.it (A.G.G.); carmelina.loguercio@unicampania.it (C.L.); alessandro.federico@unicampania.it (A.F.); 2Department of Medicine and Surgery, University of Salerno, Via Allende, 84081 Baronissi, Italy; mmasarone@unisa.it (M.M.); mpersico@unisa.it (M.P.); 3Department of Health Sciences, University Magna Graecia, viale Europa, 88100 Catanzaro, Italy; larussa@unicz.it (T.L.); l.abenavoli@unicz.it (L.A.)

**Keywords:** metabolic (dysfunction) associated fatty liver disease, nutrigenomics, nutrigenetics, diet, trained immunity, precision medicine

## Abstract

Metabolic- (dysfunction) associated fatty liver disease (MAFLD) represents the predominant hepatopathy and one of the most important systemic, metabolic-related disorders all over the world associated with severe medical and socio-economic repercussions due to its growing prevalence, clinical course (steatohepatitis and/or hepatocellular-carcinoma), and related extra-hepatic comorbidities. To date, no specific medications for the treatment of this condition exist, and the most valid recommendation for patients remains lifestyle change. MAFLD has been associated with metabolic syndrome; its development and progression are widely influenced by the interplay between genetic, environmental, and nutritional factors. Nutrigenetics and nutrigenomics findings suggest nutrition’s capability, by acting on the individual genetic background and modifying the specific epigenetic expression as well, to influence patients’ clinical outcome. Besides, immunity response is emerging as pivotal in this multifactorial scenario, suggesting the interaction between diet, genetics, and immunity as another tangled network that needs to be explored. The present review describes the genetic background contribution to MAFLD onset and worsening, its possibility to be influenced by nutritional habits, and the interplay between nutrients and immunity as one of the most promising research fields of the future in this context.

## 1. Introduction

Non-alcoholic fatty liver disease (NAFLD) represents one of the most important metabolic-related disorders of the 21st century and the leading cause of chronic liver disease and liver transplantation worldwide [1]. It includes a wide spectrum of conditions ranging from simple steatosis to steatohepatitis (NASH), characterized by the histologic appearance of inflammation and fibrosis, which act as driving factors to fuel the disease progression and complications onset [2].

Currently, NAFLD is considered the hepatic manifestation of metabolic syndrome (MS) [3] and recently, an expert consensus established that metabolic- (dysfunction) associated fatty liver disease (MAFLD) represents the more adequate denomination, revealing its larger and deeper nature as systemic disorder [4] (Figure 1).

Alarmingly, as the diffusion of MS-related conditions continues to increase and the obesity pandemic spreads irrepressibly, MAFLD prevalence, in turn, seems to rise exponentially [1]. The worldwide prevalence ranges from 6% to 35%, with higher levels in the industrialized countries (the Middle East 32%, South America 31%, United States 24%, and Europe 23%) and lower levels in the underdeveloped ones (14%) [5]. Moreover, future perspectives appear decidedly not encouraging, expecting prevalence to reach about 100 million in the United States alone by 2030 [1].

The numbers of the MAFLD spread worldwide raise several important concerns principally due to the very complex and still not fully understood pathogenesis. It gives to the disease an inherent difficulty to support the burden of its optimal medical and social management.

Pathogenetically MAFLD represents a multifactorial disease in which various elements can simultaneously contribute to the genesis and affect natural history, contributing to make its manifestation from patient to patient hugely different [6]. The genome-wide association studies (GWAS) shed light on the genetic susceptibility of the population to MAFLD onset and evolution, identifying several single nucleotide polymorphisms (SNPs), like phospholipase domain-containing protein-3 (PNPLA3) rs738409, the transmembrane 6 superfamily member 2 protein (TM6SF2) rs58542926, and membrane-bound O-acyltransferase domain containing 7 (MBOAT7) rs641738, as crucial in this scenario [7,8]. At the same time, the contemporary involvement of several environmental factors like sedentary life, unhealthy diet regimens promoting insulin resistance (IR), and gut dysbiosis, together with some immune disturbances, have been accepted by the scientific community as “tiles of MAFLD pathogenetic mosaic” [9,10,11,12] (Figure 1).

To date, no specific medications for MAFLD treatment exist [5,13]; hence, given its demonstrated capability to induce huge improvement in IR and liver damage, lifestyle modification remains the most valid and accepted recommendation [14]. This approach classically rests on three enchained pivots: regular physical exercise, weight loss, and healthy diet [15]. Regarding the latter, the scientific evidence suggests the possibility that a regimen characterized by the main consumption of plant-based food, fish, and white meat could contribute significantly to the reduction of several chronic diseases’ occurrence, including MS components and thus MAFLD [16,17,18]. These principles constitute the paradigm of the well-known Mediterranean diet (MD), which, based on the antioxidant and anti-inflammatory properties of the recommended food, currently represents the nutritional gold standard of preventive medicine [19].

However, the different kinds of therapeutic outcomes potentially obtained from this dietary regimen induced the scientific community to focus attention on the identification of the factors involved in determining the dietary therapeutic effect, hypothesizing a mutual relationship between diet and genetics [15]. Recent findings suggested nutrition’s capability, by acting on the individual genetic background and modifying the specific epigenetic expression as well, to influence NAFLD patients’ clinical severity and response to the treatment [20]. The study of the interaction between nutrients and inherited factors constitutes the central aim of emerging and promising lines of research known as nutrigenetics and nutrigenomics [21].

Globally, the main scope of these research fields is strictly linked in a very complex scientific network with other disciplines, such as genomics, transcriptomics, proteomics, metabolomics, and system biology, configuring the “Omics-approach” to the disease that seems to be the key for the full comprehension of MAFLD pathogenesis and evolution [21,22] (Figure 2).

However, further efforts will have to be spent to apply concretely the research findings in routine clinical practice and also to design, in the next future, a tailored therapeutic approach fit with the era of precision medicine.

This review describes the genetic background contribution to MAFLD onset and worsening, its possibility to be influenced by nutritional habits, and the effect of genetics in determining certain biologic responses to nutrients based on the current nutrigenetics and nutrigenomics scientific knowledge concerning metabolic-associated disorders.

## 2. Genetics and Epigenetics: An Overview on MAFLD Genetic Background

In MAFLD context, heritability seems to play a pivotal role as suggested clearly just considering the higher disease risk of offspring in case of positive family history for MAFLD, particularly when both parents are affected [23,24]. Furthermore, the ethnic susceptibility represents another overwhelming proof. Different ethnicities, in fact, show different propensity towards disease development and progression [25,26]. Interestingly, besides the inter-ethnic one, growing evidence reveals also an inter-individual variability. Therefore, people belonging to the same ethnicity have different possibilities to develop MAFLD, and this appears strictly linked to certain genetic variants as well as genetic expressions able to influence its onset and evolution [27,28,29].

### 2.1. Most Common Genetic Determinants of MAFLD

A long list of genetic inter-individual variants has been provided by GWAS in this context; it embraces several genes known as genetic determinants of MAFLD, whose expression is involved in the formation of lipid droplets and regulation of oxidative stress, inflammation, fibrogenesis, and many other metabolic pathways [7,29,30] (Table 1).

However, more robust and reproducible associations seem to exist, above all, for PNPLA3, MBOAT7, and TM6SF2 variants that currently are considered the main genetic MAFLD determinants [8].

The rs738409 SNP of the PNPLA3 gene (or Adiponutrin), encoding for the I148M protein variant, showed heavy impact on disease susceptibility. This variant has been associated with higher hepatic fat content in an IR-independent manner, as demonstrated by Romeo et al. [31]. In details, the role of IR in the light of the current scientific knowledge regarding this purpose seems to be extremely complex and, from a certain point of view, still not fully understood. In fact, the PNPLA3 gene encodes for the homonymous transmembrane protein localized on the endoplasmic reticulum [32], mainly expressed in hepatocytes and adipose tissue and exhibiting triglycerides (TG) hydrolase activity regulated by glucose and insulin [32,33]. In case of IR and obesity, both conditions featured by high insulin levels, PNPLA3 expression is induced, and the protein is placed and accumulated on the surface of lipid droplets where it is not catalytically active, disrupting TG and phospholipids turnover and remodelling [34].

PNPLA3 is also involved in the release of the storage form of retinol, known as retinyl-palmitate, in hepatic stellate cells (HSCs) [35,36]. Due to the involvement of retinyl-palmitate in the regulation of cellular fat metabolism and HSCs activation, its retention caused by the I148M genetic variant determines, consequently, the activation of pro-inflammatory and pro-fibrotic responses [35,37]. About that, growing evidence in animal and clinical experimental models suggests the I148M genetic variant relationship with all the steps of the diseases’ natural history, from simple steatosis to NASH, cirrhosis, and hepatocellular carcinoma (HCC) development [38,39]. Moreover, this SNP appears to be involved as the key driver in chronic kidney disease (CKD) development and is currently recognized as an established marker of higher cardiovascular risk in MAFLD [40].

MBOAT7 is a gene encoding for a membrane-bound enzyme whose main function is the incorporation of arachidonic acid (AA) and other unsaturated fatty acids (UFAs) in the phosphatidylinositol (PI) molecule by the Lands cycle (a series of phospholipid-remodelling reactions by which acyl-chains become trans acylated) [41]. In the presence of the rs641738 variant, the expression of the encoded enzyme decreases together with the hepatic levels of PI containing AA and causes a disruption of several cellular pathways regulating TG metabolism, inflammation, fibrosis, and cellular proliferation as well [42,43,44].

The TM6SF2 gene is responsible for the production of a protein mainly localized in the endoplasmic reticulum and Golgi apparatus, involved in the regulation of the hepatic triglyceride secretion [45,46]. The rs58542926 C>T encoding for the E167K variant of this gene is involved in MAFLD development and worsening of histological picture, impairing inflammation, ballooning, and fibrosis, as demonstrated by Liu et al. [42,47]. On the other side, this SNP seems to confer protection from the cardiovascular disease risk through the reduction of lipid secretion and very-low-density lipoproteins (VLDL) synthesis, designing a very complex biologic role that still remains to be fully clarified [48,49,50].

Besides PNPLA3, MBOAT7, and TM6SF2 SNPs, other genetic variants appear potentially involved in this scenario. Strong scientific evidence exists for the glucokinase regulator (GCKR) common missense variants rs1260326 encoding for the P446L protein [51]. The common missense variant rs1260326 disrupts GCKR function, making it unable to inhibit the glucokinase and, consequently, activating hepatic glucose uptake and glycolysis. These phenomena lead to the generation of acetyl-CoA cellular overload and de novo lipogenesis (DNL) [51]. Contrariwise, the protein phosphatase 1 regulatory subunit 3B (PPP1R3B) rs4841132 variant, through the reduction of lipogenesis and the increase of glycogen synthesis, seems to protect against hepatic fat accumulation [52]. Besides the aforementioned one involved in steatosis development, other genetic polymorphisms have been associated with the disease progression.

As known, reactive oxygen species (ROS) overproduction derived from free fatty acids (FFAs) overload in mitochondria secondary to IR and consequent mitochondrial dysfunction, represent critical events in MAFLD worsening [53]. In this regard, some studies highlighted the association of polymorphism rs4880 of Superoxide dismutase 2 (SOD 2) with higher oxidative stress levels, inflammation, and more advanced fibrosis [54,55]. In line with this, the uncoupling protein 3 (UCP3) rs1800849, uncoupling protein 2 (UCP2) rs695366, homeostatic iron regulator (HFE) C282Y rs1800562, and a rare missense variant (A165T) in mitochondrial amidoxime reducing component 1 (MARC1) represent other new interesting topics [7,56,57]. UCP3 is a mitochondrial anion carrier selectively expressed in skeletal muscle involved in the modulation of energy, lipid homeostasis, and thermogenesis by facilitating the proton leak of the mitochondrial inner membrane and uncoupling the oxidative phosphorylation [58]. Interestingly, a Spanish study conducted on a cohort of overweight patients revealed the association of UCP3 rs1800849 variant with IR, increased adiponectin levels, and NASH [59]. On the contrary, the presence of the UCP2 rs695366 variant has been linked to higher gene expression, insulin sensitivity, and protection from liver damage [60].

Other rare genetic variants were demonstrated related to MAFLD clinical context as mainly involved in the disease progression instead of its development: the rs236918 genetic variant in proprotein convertase subtilisin/Kexin type 7 (PCSK7) [61], or protecting from the disease and complication onset: the loss of function in HSD17B13 gene due to rs72613567 variant [62,63].

However, one of the most intriguing questions is how far we are to apply in the routine clinical practice the knowledge derived from GWAS studies. This question could be interpreted from some different point of views on MAFLD clinical picture: predictive, prognostic, and therapeutic ones [28]. On this line, it is necessary to highlight the lack of scientific agreement regarding the best methodological choice to evaluate the utility of genetic variant risk estimates [64]. Considering the PNPLA3 I148M variant, it has great power in the prediction of the disease, which was highlighted by several clinical trials in which its role in disease appearance was totally demonstrated independent from the other classical risk factors [24,65]. However, its pertinence as a heritable factor for NAFLD development was not demonstrated as brilliant as the latter purpose and, for this reason, the actual clinical management guidelines of the European Association for the Study of the Liver do not recommend its routine assessment for NAFLD-related liver damage evaluation [66]. In this regard, because of the lack of sufficient scientific proof to support the use of a single SNP in the prediction of the disease risk, the recent scientific literature has focused attention on the possible applicability of polygenic risk scores (PRSs) for this purpose [64]. This acquires even more relevance considering the accuracy of PRS based on well-established SNP and commonly recognized risk factors for the disease development in NAFLD risk prediction [67,68]. The use of PRS in a cross-sectional study on NAFLD was demonstrated able to induce an improvement of risk prediction in about 20% of patients. Moreover, recently, PNPLA3-TM6SF2-GCKR-MBOAT7 variants combined in a hepatic fat PRS (PRS-HFC) and adjusted for HSD17B13 (PRS-5) were demonstrated as able to predict HCC more efficiently than single variants assessment and that the association between PRS and HCC, mediated by severe fibrosis, was independent from the latter in clinically relevant subgroups and in those without advanced stages fibrosis [69]. Despite that the emerging research topic could have huge scientific impact on future clinical management, the amount of scientific evidence currently remains the most important concern to recommend their routine use. Moreover, regarding the prediction of long-term outcome, independently from the baseline staging of the disease, nowadays, it is not possible to state scientifically coherent conclusions, at least until the publication of data from long-term prospective studies.

The possibility of MAFLD therapeutic outcome prediction in the era of the patients-tailored approach assumes a fascinating aura in the light of some research trials on new therapeutic agents for disease treatment [70]. This novel research line could give the possibility to interpret better the usefulness of a wide range of already known drugs and newly developed ones as well. The WELCOME trial evaluated the response to 4 × 1000-mg capsules of 460 mg eicosapentaenoic acid and 380 mg docosahexaenoic acid administration for 15–18 months on liver fat content and fibrosis in NAFLD patients [71]. At the end of treatment, those patients carrying the 148I/I and 148I/M genotype showed a decrease of liver fat percentage (148I/I: −7.05%, 148I/M: −7.30%) whereas the 148M/M group showed a moderate increase (2.75%) [71].

Even if for some of the identified genes more powerful scientific evidences are still missing and the results of the different studies appear sometimes controversial as well, there is no doubt in considering this field one of the most promising topics of the recent MAFLD research.

### 2.2. Main Epigenetic Mechanisms of MAFLD

As part of the complex genetic background sustaining MAFLD, several epigenetic phenomena, influencing different levels of gene expression regulation, seem to be potentially involved in pathogenesis and clinical history. The mechanisms described encompass DNA methylation, histone modifications, and microRNAs (miRNAs) activity on specific targets [72]. In this context, the development of a characteristic methylation pattern could be critical and fuel the disease progression [73]. Kitamoto et al. compared the levels of DNA methylation of certain CpG islands as CpG99 (which resides in the regulatory region of PNPLA3) and CpG26 (which resides in that of PARVB variant 1) in the livers of patients with mild (fibrosis stages 0 and 1) or advanced (fibrosis stages 2 to 4) steatosis by performing targeted-bisulfite sequencing [74]. Relevantly, in the livers of patients with advanced disease, CpG26 resulted markedly hypomethylated while CpG99 was substantially hypermethylated, suggesting the hypomethylation of CpG26 and the hypermethylation of CpG99 as potential contributors to the severity of fibrosis in patients with MAFLD [74]. Moreover, in individuals affected by severe steatosis compared to mild ones, lower DNA methylation levels characterized specific CpG islands of noted pro-fibrotic genes, such as transforming growth factor- β(TGF-β), collagen 1A1, platelet-derived growth factor- α(PDGF-α), and others [75,76]. On the contrary, hypermethylation status occurred for certain CpG islands in various anti-steatotic and anti-fibrotic genes, such as ApoB and peroxisome proliferator-activated receptor (PPAR)-α [75,77,78]. The loss of PPAR-α functioning seems extremely important because it regulates cytokine production, reducing the expression of pro-inflammatory ones, and, on the other hand, it modulates the proteins involved in the fatty acid binding activity, taking part in the regulation of lipogenesis during the oxidation [77,79].

In this sense, a large number of liver DNA methylation alterations widely affecting disease onset and worsening by promoting inflammation and fibrosis has been related to certain metabolic features in terms of insulin, amino acids, and lipids serum levels. An interesting analysis of the DNA methylation pattern performed on the liver biopsies obtained from 95 obese individuals (34 individuals showing normal liver phenotype, 35 simple steatosis, and 26 steatohepatitis), identified 1292 CpG sites representing 677 unique genes differentially methylated in the livers of individuals with advanced disease (i.e., steatohepatitis) [80]. Focusing on the top-ranking 30 and another 37 CpG sites mapped to genes enriched in pathways of metabolism and cancer all together, the authors revealed 59 steatohepatitis-associated CpG sites correlating with fasting insulin levels independently of age, fasting glucose, or diabetes mellitus type 2 [80].

In addition, in a recent study on 194 obese patients (79 with liver histology classifiable as normal liver, 40 as simple steatosis, and 45 as steatohepatitis) aiming to assess serum aromatic and branched-chain amino acids levels association with steatohepatitis, the tryptophan resulted significantly higher in those with advanced disease compared to those with simple steatosis [81]. Relevantly, these amino acid serum levels result were associated with liver DNA methylation of CpG sites known to be differentially methylated in individuals with steatohepatitis and were correlated positively with serum total and low-density lipoprotein (LDL) cholesterol and, accordingly, with liver low-density lipoprotein receptor (LDL-R) at mRNA-expression level [81].

Histone modifications represent another epigenetic phenomenon potentially related to different metabolic dysfunctions [72]. The disruption of circadian rhythm of histone acetylation regulated by histone deacetylases (HDCA3), implicated in the regulation of the circadian rhythm of hepatic lipogenesis [82], may alter hepatic lipid metabolism leading to IR and obesity [83,84]. In particular, a group of deacetylates, known as silent information regulator 2 proteins (Sirtuins), may exert a key role in MAFLD development [84], according to growing evidence revealing their downregulation in animals and in vitro models [84,85]. In mice, in fact, liver-specific deletion of SIRT1 by impairing of PPAR-α signalling and decreasing fatty acids (FAs) beta-oxidation, leads to IR and related inflammation [85]. Relevantly, both the DNA methylation and histone modifications may also occur in mitochondrial DNA (mt-DNA) [86].

This feature appears in line with the aforementioned mitochondrial dysfunction implicated in MAFLD pathogenesis. Hypermethylation of mitochondrially encoded NADH ubiquinone oxidoreductase core subunit 6 (MT-ND6) was demonstrated on liver biopsy samples from NASH patients compared with subjects affected by liver steatosis [87]. In addition, the authors highlighted the association of MT-ND6 methylated/unmethylated DNA ratio with the NAFLD activity score (NAS) [87]. In keeping, a more recent study by Pirola et al. revealed higher levels of MT-Cytochrome B variance and mt-DNA damage in NASH patients compared to simple steatosis ones [86].

Regarding mt-DNA histone modifications, genetic polymorphism of SIRT3, a mitochondrial sirtuin pivotal for guaranteeing mitochondrial integrity and metabolism during oxidative stress [88], has been associated with the development of MS in mice and humans as well as contributes to downregulate autophagy, leading to lipotoxicity in hepatocytes and thus MAFLD worsening [89,90,91].

The role of epigenetics is not only limited to the DNA access variations being expressed also in post-transcriptional steps acting by miRNAs, small non-coding single strand RNAs (ssRNAs) able to repress or degrade specific mRNAs target, regulate several biological and pathological processes, and intervene in lipid metabolism and inflammatory processes [92,93]. In this context, one of the most relevant studied miRNAs is miRNA-122, identified as a promising biomarker and drug target for MAFLD [94]. Esau et al. firstly highlighted the key role of this miRNA in the liver of high-fat diet (HFD) fed mice; after treatment with antisense oligonucleotides (ASO) inhibitors of miRNA-122, a significant reduction of hepatic biosynthesis of FAs, an increased beta-oxidation, and a reduction of TG accumulation were observed [95]. Recent scientific evidence demonstrated through in vitro and animal models the miRNA-122 suppressing role of SIRT1 expression via binding its 3’-untranslated region (UTR), enhancing lipogenesis, and contributing, thus, to lipid accumulation [94].

In contrast, miRNA-122 knockdown mitigated this consequence through the upregulation of SIRT1 and the activation of the liver kinase B1 (LKB1)/adenosine monophosphate-activated protein kinase (AMPK) signalling pathway [94]. Besides miRNA-122, other miRNAs could interfere with these processes [96]: for instance, miRNA-33a/b action on the target AMPK, by reducing AMPK expression, increases the levels of intrahepatic TG [97]; contrariwise, miRNA-33a/b inhibition promotes beta-oxidation of FAs and insulin sensitivity [97,98]. However, the role of miRNAs was not only related to the induction of steatosis because of the proved effects on the progression to inflammation and fibrosis [99]. In this sense, miRNA-34a results highly expressed in MAFLD patients in a stage-dependent manner [100,101,102]. On the contrary, the expression levels of miRNA-451, able to downregulate nuclear factor kappa-light-chain-enhancer of activated B cells (NF-κB) and tumour necrosis factor–α(TNF-α) in vitro, resulted significantly decreased in NASH patients [101].

Altogether, these epigenetic mechanisms contribute to MAFLD inter-individual variability in terms of susceptibility towards disease and its progression, influencing the clinical history of each patient. To improve MAFLD individual prognosis, the future challenge is certainly represented by the application of these findings to the routine clinical practice, through the identification of biomarkers and therapeutic targets usable for early diagnosis and personalized therapies. In these terms, the above-presented findings highlight a suggestive relationship between the occurrence of epigenetic phenomena modifying gene expression and metabolic dysfunctions (including, among others, IR and high LDL serum levels). This feature appears widely relevant, representing an important food for thought regarding potential nutrigenomics approaches considering certain nutrients’ capability to influence gene expression, discussed as well in the next paragraph.

## 3. Nutrigenomics in MAFLD: The Impact of Nutrients on Epigenetic Regulation

The integration of the Omics approach and system biology for MAFLD study certainly represents one of the most important novelties of the recent scientific literature. Thanks to an increasing number of high-throughput Omics investigations published in the last few years, the molecular pathobiology of MAFLD, considered as a systemic disorder, has been better clarified. This fact represented an important starting point to properly debate regarding the appropriacy of the holistic view on MAFLD pathophysiology, potentially leading to some important diagnostic and therapeutic advantages as well [103]. As part of this tangled biologic network, nutrigenomics appears to be crucial in this setting, particularly considering the potential benefit that could derive from the dietary therapeutic application and tailored clinical management. In this sense, nutrigenomics represents a part of the well-known environment exposition/disease development relationship. The origin of this research field is based on the observation of a gene-diet effect due to a drastic change in the exposition to specific macronutrients, such as lipids, carbohydrates, or proteins, as well as considering the availability of micronutrients, involved as cofactors in DNA restructuring. In fact, it seems to be able to influence the expression and thus the activity of several enzymes involved in nutrients’ metabolism conducing, potentially, to a metabolic imbalance [104].

### 3.1. Lipids

The food enrichment with omega 6 (n-6) 18-carbon polyunsaturated fatty acids (PUFAs), in particular linoleic acid (LA), linked to the alteration of LA/alpha-linolenic acid ratio, represents a classic example of the latter concept, conducing to an altered omega 3 (n-3) and n-6 metabolism of inflammatory PUFAs and enhancing, then, the production of inflammatory mediators involved in several disorders [105,106]. The DNL represents, together with the adipose tissue fat release and the block of beta-oxidation, a crucial mechanism able to trigger the hepatocyte TG overload induced by high short-chain fatty acids (SCFAs) consumption. They have been hypothesized to exert this effect, inducing the activation of the peroxisome proliferator-activated receptor (PPAR)-coactivator-1 beta (PGC-1) and enhancing the expression of stearoyl-CoA desaturase-1, fatty acid synthase, diacylglycerol acyltransferase genes involved in lipid metabolism and homeostasis [107]. The clinical effect of high n-6/n-3 PUFAs ratio and SCFAs-rich food consumption is represented by both hepatocyte and lipocyte dysfunction that could lead to cellular fat accumulation, inflammation, and fibrosis in a typical NASH histologic picture [19,53].

Contrariwise, the increase of n-3 PUFAs in the daily diet was demonstrated in preclinical models as related to suppression of diet-induced steatosis and improvement of IR, mainly reducing the activity of sterol regulatory element-binding proteins (SREBP)-1 transcription factor, the expression of DNL, and fat uptake genes, as well as by increasing the expression of genes involved in FAs oxidation and export thought VLDL secretion [108,109]. These gene-diet molecular mechanisms of interaction lead also to important consequences on the major cause of death in this context: the cardiovascular events. In fact, it was well established that n-3-enriched food consumption or a low n-6/n-3 ratio is linked to an improvement of the blood lipid profile, lowering the LDL level up to 20 % along with glycemia and IR and reducing the risk of death for coronary heart diseases [110,111]. However, as recently highlighted in the FADSDIET and FADSDIET2 trials, an interconnection between the fatty acid desaturase (FADS) rs174550 gene cluster and dietary linoleic and alpha-linolenic acid intake on plasma lipids composition, fasting glucose, and high-sensitivity C-reactive protein (hsCRP) exists, in particular for TT homozygous carriers in which plasma eicosanoid concentrations correlated with the arachidonic acid proportion in plasma and with hsCRP [112,113].

The effect of diet in the explored context embraces also the miRNA pathobiology and, on this purpose, Long et al. highlighted a SIRT1-suppressing effect exerted by FFAs and a high-fat diet able to promote lipogenesis in HepG2, HuH7 cell lines, and mice thought miR-122 upregulation [94]. MiR-181b and miR34a have been recognized to play crucial roles in steatosis development, acting as SIRT1 suppressors in case of palmitic acid exposure and both SIRT1 and PPAR suppressors in case of HFD exposure [114,115], as well as inhibiting β-Klotho, a co-receptor of the fibroblast growth factor 19 (FGF19), mainly involved in glucose metabolism, whose reduction was demonstrated related to inflammation, ballooning, and fibrosis [116].

### 3.2. Carbohydrates

Regarding carbohydrates, powerful evidence for fructose exposition exists. In fact, fructose-enriched food has been recognized as one of the most important categories of food directly involved in MAFLD onset and evolution [117]. The pleiotropic role of fructose in this setting seems to be exerted though a direct effect as substrate for DNL, increasing the acetil-CoA cellular level and acting, in a relative IR independent-manner, as an enhancer for the expression of several enzymes involved in DNL via SREBP-1c activation [118,119]. The described molecular mechanisms are involved in determining clinical outcomes for MAFLD patients and, on this purpose, the fructose-enriched drink consumption was directly related to the risk of hepatic steatosis, inflammation, and fibrosis development as well as related to extrahepatic complications onset [120,121] even if it has to be note that the highlighted effects appear to have a powerful scientific background only in case of concomitant hypercaloric dietary regimens in clinical settings [122,123].

### 3.3. Proteins

Differently to the previous categories of macronutrients, the role of dietary proteins in this context appears still obscure and, for certain points of view, controversial due to the low availability of data coming from well-constructed controlled trials that, thus, make it impossible to deduce scientifically coherent conclusions. In order to address the main scope of this review, we believe it correct to pay attention to the association between low dietary protein intake and the risk of sarcopenia development. In fact, sarcopenia and sarcopenic obesity properly represent well known risk factors for MAFLD progression, fueling IR, hepatic inflammation, and fibrosis as well as atherogenesis [124]. Even if the molecular mechanisms that lead to the development of sarcopenia and sarcopenic obesity are not fully clarified yet, it is not questionable that diet, physical exercise, and aging represent the most important factors of the pathogenetic cascade. If then, sarcopenia is able to have a worsened effect on the MAFLD clinical picture, is it possible to hypothesize an effect exerted by the muscle loss on the expression of genes involved in metabolic homeostasis? In this line, the role of autophagy in determining the metabolic imbalance able to trigger fat deposition and fueling muscle loss in a vicious circle seems to be crucial [125]. The generation of the autophagosome for the degradation of misfolded proteins and cellular organelles represents an efficient strategy of energy recovery and re-allocation to other biologic pathways [126]. However, the hyperactivation of this process, in particular when it occurs in mitochondria, determine their dysfunction, impairing the metabolism of the whole cell and then the organ physiology [127]. Higher levels of autophagy and mitophagy have been hypothesized to be important determinants for sarcopenia and reduced physical performance [128,129]. Sarcopenia, in turn, was related to increase in the expression of several genes involved in autophagy regulation, such as p62, autophagy related 7 (ATG7), beclin-1, and p53 [125]. Moreover, a crucial role of sarcopenia in inducing the upregulation of PPAR-α, SREBP-1, and acetyl-CoA carboxylase (ACC) was also demonstrated and related to the increase of fatty acid synthesis responsible for muscle cellular fat accumulation [125].

### 3.4. Micronutrients

As previously mentioned, the availability of several micronutrients could influence the DNA methylation state, regulating, then, the expression of crucial genes involved in metabolism control. It is necessary to highlight that the scientific knowledge produced about this topic are mainly based on animal experiments, and due to the lack of well-designed clinical trials, some controversies about human results exist. However, it has been shown that betaine, choline, vitamin B12, and folate could be related to MAFLD progression and their supplementation in a high-fat-sucrose- (HFS) fed mice model, acting on fatty acid synthetase (FASN) hypermethylation, improved the liver TG overload [130]. Moreover, betaine has been correlated with hypomethylation of CpG clusters of microsomal triglyceride transfer protein (MTTP), enhancing VLDL excretion from the liver; conversely, vitamin B12 and folate deficiency was demonstrated as related to mitochondrial dysfunction, impaired beta-oxidation, and TG accumulation based on peroxisome proliferator-activated receptor gamma coactivator 1-alpha hypomethylation [131].

### 3.5. On the Way of a Genetic Based Dietary Approach

Thanks to the actual scientific preclinical evidence, it could be reasonable to hypothesize the possibility of a genetic-based nutritional recommendation for a tailored MAFLD therapeutic management. The improvement of fatty acids oxidation, glutathione, and NAD+ availability may provide interesting strategies in this context [132].

Moreover, in a murine model, the supplementation with betaine, choline, cobalamin (vitamin B12), and folic acid in a high-fat sucrose diet regimen determined the prevention of fat accumulation through the epigenetic modulation of several genes involved in insulin sensitivity and globally lipid metabolism homeostasis [133].

Previously, highlighted in 154 NAFLD patients was also a PNPLA3 involvement in the regulation of the therapeutic response to 30 min per day of physical exercise and an individual-designed diet regimen based on the consumption of fruit, vegetables, moderate-carbohydrate, low-fat, low-glycaemic index, and low-caloric products [134].

G-allele carriers demonstrated a greater reduction of intrahepatic triglyceride content compared to those with the C-allele after 12 months of treatment together with an improvement of body weight, WHtR, total cholesterol, and LDL [134].

Due to the novelty of the topic and the weak, even if hugely promising, scientific evidence, it is not possible to transfer the results of the studies in the routine clinical practice as well as compare the therapeutic outcome of conventional general healthy diet recommendation to a genetic-based approach. Nevertheless, the charm and the power of the latter purpose represents the reason why we observe an exponential rise in the number of scientific studies on that topic and the increase of direct-to-consumer, genetically-based nutritional testing and advice by nutritional companies [135]. Globally, these signals have to be recognized as a measure of the scientific interest and expectation that society shows regarding this topic that may derive from the comprehension of the MAFLD problem worldwide not only among specific categories of healthcare professionals but also in the general population.

## 4. Nutrigenetics in MAFLD: Genetics Influence the Response to the Nutrients

A plethora of genetic polymorphisms contribute to the determination of a certain background influencing MAFLD by interfering with different mechanisms involved in several above-mentioned metabolic pathways [7]. In addition to these features, the genetics involvement in affecting the human biologic response to the nutrients (in terms of metabolism, requirement, and food tolerances) has been recently emerging [136]. This latter concept constitutes the study field of nutrigenetics and appears crucial in the genesis and progression of several diseases [21,137]. Consistently with the definition of MAFLD as a multifactorial disorder, the exposition to certain nutrients represents a recognized environmental factor whose interaction with a specific genetic background may lead to different clinical consequences. Paradoxically, in this scenario, a specific kind of food may be hazardous for some MAFLD patients and protective for others, depending on the presence of specific genetic variants of PNPLA3, MBOAT7, TM6SF2, and GKCR (Figure 3) [21,137].

Figure 3 proposes the mechanisms by which the genetic determinants of MAFLD influence the response to nutrients. The PNPLA3 I148M protein variant reduces the hydrolysing of n-9 PUFAs. As a consequence, the intake of foods containing n-3 PUFAs (omega-3) (above all, fishes such as salmon and tuna) ameliorates the hepatic steatosis; conversely, the assumption of n-6 (omega-6) PUFAs (present in palm oil) or n-9 derived PUFAs (contained among others in nutrients as red meat, olive oil, and almonds), through the n-6 overloading, increases the TG synthesis and promotes the production of omega-6 pro-inflammatory-derived species (as AA). Moreover, the omega-3 may downregulate SREBP1 activity while the carbohydrate feeding (particularly sugar) may up-regulate it.

The presence of TM6SF2 E167K protein variant has two apparently opposite effects: on one side, it entails lipid retention (both in hepatocytes and in enterocytes); on the other, by worsening insulin resistance and reducing VLDL secretion from liver cells, it increases hepatic steatosis. These latter effects are highlighted after high carbohydrate assumption or in case of high-fat diet regimen. Besides, in the presence of GCKR P446L protein variant, a high carbohydrate assumption may increase hepatic FAs accumulation and thus steatosis. This latter, in the presence of MBOAT7 rs641738 C>T variant, is most favoured by hyperinsulinemia, as well as the assumption of saturated FAs.

### 4.1. PNPLA3 rs738409 Variant’s Influence

The rs738409 C>G variant in PNPLA3 influences the response to the dietary introduction of PUFAs, particularly omega 9 (n-9) [138,139]. The n-9 PUFAs derive from n-6 PUFAs or directly from the diet (being present in meat, almonds, and several other components of the MD, such as olive oil) [140]. In case of I148M protein variant, the reduction of PNPLA3 hydrolytic efficiency decreases the hydrolysing of n-9 fatty PUFAs, leading to an n-6 overload and two critical consequences: the TG synthesis with increased hepatocyte overload and the production of n-6 pro-inflammatory-derived species (e.g., AA), worsening the steatosis and promoting NASH [141,142] (Figure 3). In contrast, in a randomized controlled trial (NCT01038102) comparing the effects of n-6 PUFAs with short FAs on liver fat and inflammation, high n-6 PUFAs intake after 10 weeks did not increase inflammation or oxidative stress [143]. Therefore, contrary to the better-known consequences on cardiovascular diseases [144], n-6 PUFAs’ effects on the liver fat content and their capability to promote inflammation still remain unclear [142,143]. Due to the controversy regarding this purpose, the construction of large clinical trials focusing attention on the genetic background based on the actual nutrigenetics and nutrigenomics knowledge could provide to the scientific community the missing key of lecture to interpret in deep the obtained results. In presence of the I149M protein variant, the consumption of n-3 PUFAs, largely contained in food as salmon and tuna, reduced TG accumulation in liver cells, thus ameliorating hepatic steatosis [21,141,145] (Figure 3). Regarding that, a previous cohort study on 127 NAFLD-obese youths’ homozygote for the G allele of the rs738409 revealed that n-3 consumption ameliorated steatosis and alanine aminotransferase (ALT) levels [141].

Moreover, in a recent randomized controlled trial (NCT01556113) on a group of twenty obese adolescents, a 12 weeks’ low n-6:n-3 PUFAs ratio (4:1) norm caloric diet mitigated fatty liver determining the reduction of ALT levels and hepatic fat fraction percentage (HFF%) assessed by multiple abdominal slices with magnetic resonance imaging [146]. Relevantly, even if the HFF% declination was observed in both CC/CG and GG PNPLA3 rs738409 genotype patients groups, it resulted significant in the GG one [146]. These findings suggest the PNPLA3 rs738409 variant capability to affect the response to this dietary intervention.

Besides PUFAs, the presence of the PNPLA3 rs738409 variant has been shown associated also with the response to carbohydrates [147]. In particular, as highlighted in the trial NCT00697580, based on a cohort of 153 Hispanic children, the carriers of the GG genotype were more susceptible to the hepatic fat overload (expressed as HFF%) when exposed to high dietary carbohydrate intake, specifically sugar [147]. Consistently, an intriguing study by Nobili et al. revealed a significant interaction between PNPLA3 I148M and the intake of sweetened beverages, showing increased hepatic fat deposition and steatosis [148]. The above-presented results appear linked to the following contrast effect: on one side, n-3 PUFAs feeding downregulates SREBP1c expression [149]; on the contrary, the carbohydrate feeding upregulates it [150]. The activation of SREBP1c/liver X receptor (LXR) pathway represents a critical moment in MAFLD development since it promotes PNPLA3 expression [150]. Relevantly, in the presence of its rs738409, C>G variant, a n-3 PUFAs-poor and carbohydrate-rich diet may increase I148M protein expression levels, thus promoting FAs overload and inhibiting, in turn, PNPLA3 degradation in a vicious circle (Figure 3).

In addition to PNPLA3, other genetic variants have been associated with a specific response to certain nutrients and dietary habits, showing important repercussions on MAFLD clinical history.

### 4.2. MBOAT7 rs641738 Variant’s Influence

MBOAT7’s main function is the conjugation of an acyl-CoA to the second acyl-chain of lysophospholipids, using as preferential substrate the arachidonoyl-CoA, regulating, then, the desaturation of phospholipids and the amount of free AA [151]. The rs641738 C>T variant of MBOAT7 has been associated with the development of steatosis and its complications [44,48].

In the presence of the T allele, as well as in the case of hyperinsulinemia, MBOAT7 expression (and function) results as impaired, promoting the formation of saturated phospholipids [152]. The accumulation of these latter compounds favours the synthesis of saturated and mono-unsaturated TG, promoting fat deposition in liver cells. As a consequence, in C>T variant carriers, the high intake of hyperglycaemic nutrients and food rich in saturated fatty acid may unmask MBOAT7 deficiency by increasing the predisposition to fatty liver disease and its advanced stages (Figure 3).

### 4.3. TM6SF2 rs58542926 and GCKR rs1260326 Variants’ Influence

The rs58542926 C>T TM6SF2 variant, encoding for the E167K protein, has been previously linked with reduced circulating lipid levels and increased hepatic steatosis due to the lipids retention (in enterocytes and hepatocytes) and the reduction of VLDL release [48,49,50]. The presence of this variant has been associated with lower post-prandial lipemia levels even after a high-fat challenge (Figure 3), as shown in a cohort of 3556 individuals enrolled in the Amish Complex Disease Research Program (ACDRP) [153]. However, besides the effects on lipid metabolism, the E167K protein variant seems to also influence the response to carbohydrate assumption and thus glucose tolerance. In line with this, Musso et al. revealed the association of TM6SF2 C>T polymorphism with increased hepatic (and adipose) IR as well as impaired pancreatic β-cell function [154]. These mechanisms, together with the abovementioned lipid retention and reduced VLDL liver secretion, may explain the TM6SF2 C>T carriers’ susceptibility towards hepatic steatosis in case of high-carbohydrate dietary regimens (Figure 3).

Finally, the common missense variant rs1260326 of GCKR gene, encoding for the P446L protein, has been shown to disrupt the regulation of glucose homeostasis by increasing glycolysis and hepatic DNL [51]. Growing evidence suggests the GCKR variant involvement to influence the response to different nutrients [155]. In this sense, Nettleton et al. revealed, in patients carrying the GCKR variant, the association between high whole-grain intake and the reduction of fasting insulin concentration [156]. Moreover, after the consumption of carbohydrate enriched (CHO) drink (75 g glucose and 25 g fructose), a group of 14 obese adolescents carriers of homozygote rs1260326 variant showed increased hepatic lipid synthesis due to the enhanced glycolytic carbon flux to TG formation [157] (Figure 3).

### 4.4. Personalized Nutritional Strategies Targeting Specific Genetic Backgrounds

Basing on the all above presented observations, a possible targeted therapy for the homozygous patient carriers of the PNPLA3 rs738409 variant may be based on the following three recommendations: (1) to increase the dietary intake of foods rich in n-3 PUFAs (such as salmon, tuna), (2) to follow a diet with a high n-3 PUFAs/ n-6 PUFAs ratio, and (3) to decrease the intake of sweetened food [148]. Differently, in the case of TM6SF2 rs58542926 C>T variant, a reduction of carbohydrate intake rather than lipids may result in more useful outcome [154]. In keeping, a diet based on low carbohydrate and saturated fatty acid consumption may affect the genetic susceptibility to develop fatty liver in carriers of the MBOAT7 rs641738 variant [21]. However, these promising objectives still need to be addressed via large randomized clinical trials from which it might be possible to look at the MAFLD clinical picture from another point of view, based, moreover, on the possibility to corroborate the unquestionable potential advantage that could derive from the use in the routine clinical practice of prognostic/predictive genetic scores as previously mentioned.

## 5. Future Perspectives: Diet, Immunity, and Genes Interaction, a Very Complex Network That Needs to Be Explored

The tangled network sustaining MAFLD pathobiology seems not only limited to the role of the caloric overload due to the dietary habit, showing extraordinary and deep interconnections among genetics, metabolism homeostasis, gut microbiome modification, and immune response modulation. In this setting, the development of westernized societies as well as low-income countries is linked to deep changes in lifestyle, characterized by sedentary life and western-type dietary regimens, which determines the irrepressible spread of immune-metabolic associated disorders, forcing governments all over the world to spend strong efforts to support the social and economic burden of disease [158,159].

Novel evidence of the recent scientific literature has been identified in the innate and adaptive immunity response triggered by and linked to FFAs-associated mitochondrial dysfunction, a crucial role in the promotion of oxidative stress, IR, and inflammation that represents the pathogenetic triad for MAFLD progression [160,161].

The close interconnection between immune reaction and cell metabolism could be considered the cornerstone for the comprehension of several immune-mediated metabolic disorders, and, in this scenario, the immunometabolic dependence of the MAFLD clinical picture seems to be compliant with the actual knowledge on this topic [10].

We are what we eat, and it is particularly true considering the interplay between dietary habits and the immune system reaction. The effect of diet composition in this context seems very wide due to its influence from cytokine production up to bone marrow immune cells generation or maturation in a specific subtype [162,163,164]. Characterized by the richness in refined sugars, salt, white flour, processed meats, purified animal fats, and food additives and the lack of fibres, vitamins, minerals, and other plant-derived molecules such as antioxidants, western diet regimens were largely explored for their possibility in influencing immune activity [165,166]. It has been well established in animal and clinical models the influence of these dietary patterns with elevated serum markers of inflammation, confirming a kind of mutual relationship between the quantity and quality of daily calories, the nutritional and adipose tissue endocrine functioning, and the immune system activity [167]. In this context, the role of cholesterol as an atherosclerosis trigger was largely investigated, acting as a key factor in macrophages’ activation and systemic inflammation induction [168,169]. Similar to cholesterol, dietary palmitate can activate the murine NLRP3 inflammasome, leading to mitochondrial dysfunction and oxidative stress generation via 5’ AMPK inhibition [170]. However, the effect of diet on immune system functioning is not only limited to a direct activity on immune cells modulating the gut microbiota composition that, in turn, influences the immune homeostasis. The diet-induced modification of gut microbiota and microbiota metabolites’ production represent the crucial events for the intestinal permeability increase, metabolic endotoxemia induction, and systemic inflammation establishment via immune scavenger receptors’ activation [171].

Conversely, dietary patterns rich in fibres, polyphenols, and natural and unprocessed food, such as in the Mediterranean dietary pattern, has been related in some clinical trials with anti-inflammatory properties, confirmed by the lowering of serum inflammatory markers, cytokines, and associated as well with reduced metabolic and cardiovascular risk, including myocardial infarction or stroke, by up to 30% [172,173]. On this line, oleate was related to the dropping of palmitate-related ER stress and apoptosis in THP-1 cells and human macrophages [174].

Omega-3 fatty acids as well can directly modulate inflammatory responses in a GPR120-dependent and -independent manner by lowering the activation of scavenger receptors like TLR2, TLR4, and tumour necrosis factor receptor [175].

Some nondigestible fibres, particularly abundant in a typical Mediterranean diet regimen, are fermented to short-chain fatty acids (SCFA): acetate, propionate, and butyrate that may act locally and systemically as immune modulators by both activating G protein-coupled receptor (GPR) 41, 43, and 109a expressed on murine intestinal cells, limiting in turn dendritic cells’ proinflammatory activity and inhibiting the differentiation of T helper 1 cells [176,177,178]. Moreover, they act directly as inducers for regulatory T cells’ differentiation and B cells’ homeostasis factors [179].

It is reasonable, thus, to hypothesize an effect derived from different dietary habits on the regulation of the immune-related genes’ expression that could fuel, together with the highlighted previously mentioned alterations of metabolic pathways, MAFLD evolution. In this context, inflammation and oxidative stress occur simultaneously, influencing each other and determining the development of the main pathogenetic determinants for the MAFLD-associated extrahepatic complications onset [180].

It has to be noticed that currently, oxidative stress represents the main determinant for adaptive immunity recruitment in MAFLD evolution, mainly triggered by oxidized phospholipids and aldehydes production linked to oxidation-specific epitopes (OSEs) generation [181,182]. It has been demonstrated a high prevalence of specific anti-OSE immunoglobulin G (IgG) production in NAFLD and NASH patients, highlighting also the direct correlation with hepatic lobular inflammation and B- and T-cell tissue infiltration [183,184].

In line with this, in preclinical models of NAFLD and NASH, it was already demonstrated the association between anti-OSE IgG titres and B-cell maturation inhibited by antioxidant treatment [183,185]. Moreover, pre-immunized malondialdehyde-acetaldehyde-adducts mice showed M1 Kupffer cells’ activation sustained by liver Th1 polarization of CD4+ T lymphocytes [183]. It was demonstrated, accordingly, a huge reduction of hepatic CD4+ regulatory T cells (Tregs) lymphocytes [186], mainly due to the induction of apoptosis, hence facilitating Th1 and Th17 proinflammatory activity [187].

Normally, the regulation of cellular cholesterol income is based on LDL-R expression on cellular surface that, in turn, undergoes an intracellular, cholesterol-level negative feedback. However, unhealthy dietary regimens characterized by high fat/sugar consumption or lifestyles such as smoking habits and low physical activity are able to determine oxidative stress imbalance. The oxidized LDL (oxLDL) are recognized by several macrophages’ scavenger receptors (CD36, scavenger receptor A- SR-A, lectin-like oxidized LDL receptor-1- LOX-1, toll-like receptors-TLRs) whose expression is not affected by intracellular cholesterol level [169,188].

OxLDL and the accumulation of crystalized cholesterol lead to macrophages’ activation, NLR family pyrin domain containing 3 (NLRP3) induction, and inflammatory cytokines genes overexpression through several different and not fully clarified mechanisms involving, in part, the recent fascinating concept of the discussed-below trained immunity (TI) pathobiology [170,188,189]. As essential component of the cellular wall, cholesterol seems to exert a dose-dependent immunomodulatory effect via LXR induction. This phenomenon, insured by a normal mitochondria biological activity, is mainly based on the generation of 25 and 27 hydroxycholesterols, physiological ligands for LXR whose activation has been correlated with anti-inflammatory effects [190]. Based on this interesting scientific evidence and due to the well-known mitochondrial dysfunction as well as the metabolic Krebs cycle MAFLD-related imbalance, the dose-dependent anti-inflammatory effects of cholesterol disappear, acquiring, then, a potentially important role in fueling hepatic inflammation [191]. SCFAs are involved in the link between dietary habits and inflammatory genetic reprogramming, acting directly on TLR4 and, consequently, triggering NF-kB or inducing intestinal permeability increase [192]. The FAs overload and adipose tissue accumulation could lead to two major consequences: altered IR-related beta-oxidation and modification of resident immune cells repertoire [193]. Normally the tissue-resident macrophages regulate the adipocytes function, maintaining the anti-inflammatory state, enhancing the production of several cytokines such as Interleukin (IL)-10, and favouring the beta-oxidation genes’ transcription [194]. In the case of chronic low-grade inflammation, a change in the specific macrophages’ subtype and their activity has been demonstrated, correlated to IL-1β, TNF-α, IL-6 genes’ transcriptions and IR worsening, generating a vicious circle able to affect hugely MAFLD [195]. Moreover, this resident immune dysfunction seems to be directly related also to the lowering of tissue-specific IL-10 producer B lymphocytes, a condition that was directly held responsible for the worsening of the main driver of liver steatosis: the IR [196]. As above described, the TI pathobiology could be another interesting and potentially useful therapeutic research field in this context [10]. OxLDL, high amounts of glucose, glycation end products, FAs, or microbiome-associated metabolites can induce an epigenetic reprogramming typically associated with a kind of innate cells’ immunological memory by which especially macrophages show a powerful inflammatory reaction potentially obtained from second aspecific antigenic contact [197]. The mammalian target of rapamycin (mTOR) pathway genes are transcriptionally upregulated in case of trained response that lasts from a few days to several months, thanks to the crosstalk between peripheral macrophages and the new cellular clone in the bone marrow that preserve the typical TI reaction despite the lack of a first antigenic contact [198]. It is, on this purpose, extremely interesting to analyse the concept of obesogenic immunological memory, hypothesized on the basis of the recent TI knowledge, to explain the lack of metabolism homeostasis recovering in case of weight loss, demonstrating also a stable, high cardiovascular risk [199,200]. Western diet, then, represents one of the most important promoters of sterile inflammation acting via NLRP3 and inducing the transcriptional rewriting of innate immune training also at the level of myeloid progenitors in bone marrow [201,202,203]. Notably, it enhances the HSCs activation in the liver, triggering inflammation and fibrogenesis as well as ensuring the progression of the disease and potentially complications onset [202,203,204]. However, even if the highlighted evidence seems to be promising because of the possible consequences in the comprehension of mechanisms sustaining MAFLD and the renewal of clinical and therapeutic management, it has to be noticed that the greater part of the actual knowledge are based on animal studies or referred to other inflammatory dysmetabolic contexts similar to but not as complex as MAFLD. Furthermore, the lack of large cohort studies or clinical trials on these topics still remains the Achilles’ heel that needs to be addressed.

## 6. Conclusions

Due to the worldwide spread of the metabolic pandemic and the epidemiological change of the modern hepatology, the need to pay attention to the mechanisms sustaining MAFLD and related disorders have become a diktat for future research perspectives. In light of recent scientific knowledge, diet and genetics establish the cornerstone of this complex pathologic picture for their involvement in all the different steps of the disease: development, worsening, complication onset, and therapy as well. The key for the deep and complete comprehension of the mechanisms underlying MAFLD could be referred to as diet-genes mutual relationship also in order to take advantage of designing tailored management approaches for the future of medicine. To achieve this goal, future, strong scientific efforts in designing randomized controlled trials aimed at the study of the interaction between diet and genetic factors, enrolling large cohorts of patients with long-term follow up, have to be spent.

## Figures and Tables

**Figure 1 nutrients-13-01679-f001:**
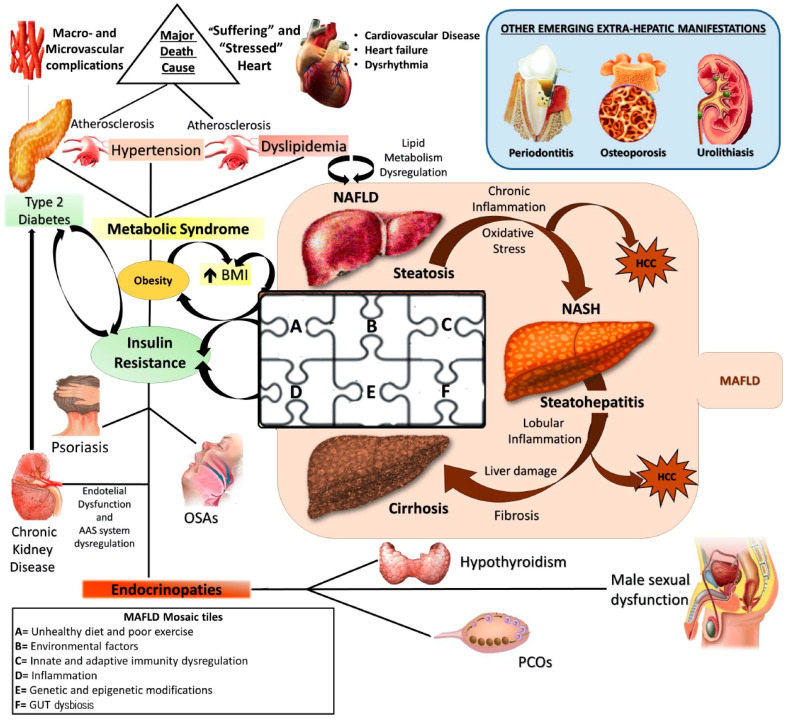
NAFLD: a metabolic systemic disease. The modern approach considers NAFLD as a metabolic, systemic disease characterized by several extra-hepatic manifestations mostly linked by a status of insulin resistance (IR). EDCs: Endocrine disrupting compounds; IR: Insulin resistance; MAFLD: metabolic (dysfunction) associated fatty liver disease; NAFLD: Non-alcoholic fatty liver disease; NASH: Non-alcoholic steatohepatitis; HCC: Hepatocellular carcinoma; OSA: Obstructive sleep apnea; PCOS: Polycystic ovarian syndrome; T2DM: Type 2 Diabetes Mellitus.

**Figure 2 nutrients-13-01679-f002:**
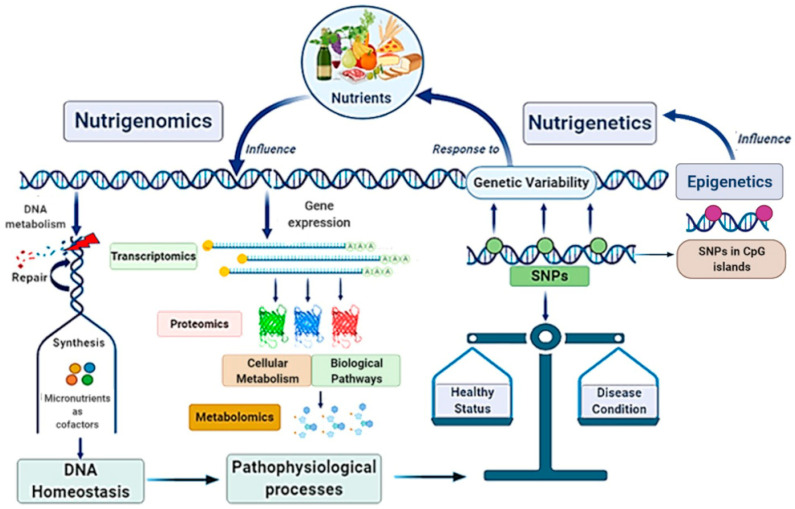
Nutrigenomics and nutrigenetics: two sides of the same medal with several linked approaches. Genome influences the responsiveness to nutrients (nutrigenetics’ field); at the same time, nutrition can also modify gene expression involving epigenetic mechanisms (nutrigenomics’ field). Nutrigenomics investigates the effects deriving from the interaction between the nutritional environment and inherited factors. Given the complexity of the scenario, nutrigenomics quests include several approaches involving many other disciplines. Nutritional factors and genetic ones influence each other: on one side, nutrients affect DNA metabolism, gene expression, and genetic variability; on the other side, genetic variants (as SNPs), by determining specific individual genotype, influence dietary habits. In turn, nutrigenetics could also be influenced by epigenetics. Altogether, these mechanisms contribute to determinate the status of health or a condition disease. SNPs, Single Nucleotide Polymorphisms.

**Figure 3 nutrients-13-01679-f003:**
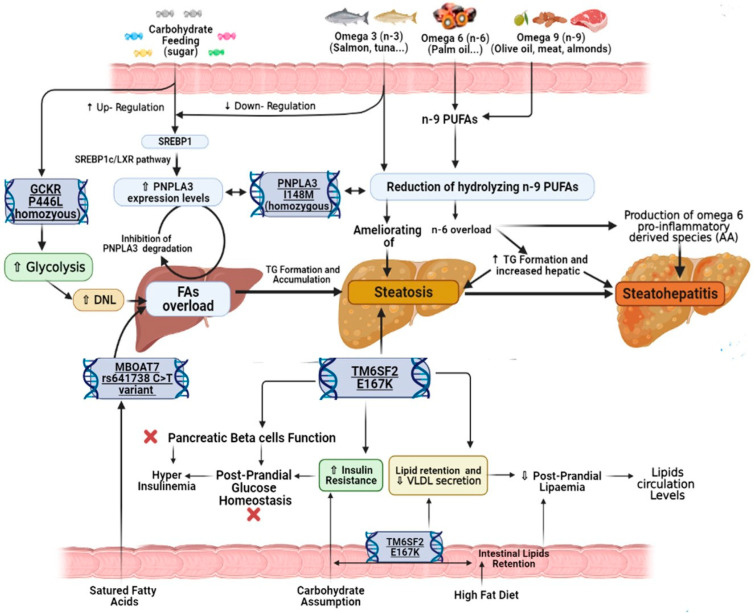
Genetic determinants of MAFLD influence response to the nutrients. *PNPLA3, Phospholipase domain-containing protein-3; MBOAT7, Membrane-bound O-acyltransferase domain containing 7; TM6SF2, Transmembrane 6 superfamily member 2 protein; GCKR, Glucokinase regulator; SREBP1, Sterol regulatory element-binding proteins;* VLD, Very-low-density lipoproteins; FAs, Fatty acids.

**Table 1 nutrients-13-01679-t001:** Main genetic determinants of NAFLD.

	Gene	Variants/SNPs/Protein Variants	Relative Effects/Association with
Major and most common genetic determinants of MAFLD	*PNPLA3*	rs738409 (I148M)	Disruption of triglycerides and phospholipids turnover and remodelling: increased hepatic fat accumulation; Disruption of retinol storage in HSCs leading to higher risk of inflammation, fibrosis, and HCC progression.
	*MBOAT 7*	rs641738	Higher risk of MAFLD development, inflammation, fibrosis, and HCC progression.
	*TM6SF2*	rs58542926	Favouring liver fat accumulation;Protection against the development of cardiovascular diseases.
Other genetic determinants involved in lipid metabolism	*GCKR*	rs1260326	Increased de novo lipogenesis and worsened hepatic steatosis.
	*PPP1R3B*	rs4841132	Reduction of de novo lipogenesis and thus protection from hepatic fat accumulation.
	*APOB*	Several and different	Reduced VLDL export from hepatocytes.
Other genetic determinants involved in oxidative stress imbalance	*SOD2*	rs4880	Higher oxidative stress and more advanced fibrosis.
	*UCP3*	rs1800849	IR worsening, increased adiponectin levels, and NASH development.
	*UCP2*	rs695366	Higher insulin sensitivity and protection against liver damage.
	*MARC1*	A165T	Lower hepatic fat accumulation and decreased levels of several biomarkers of liver disease.
	*HFE*	rs1800562 (C282Y)	Iron overload and related oxidative stress imbalance.
Other genetic determinants involved in inflammation and fibrosis	*TLR4*	D299G and T399I	Protection against fibrosis (in animal models).
	*IFNL4*	rs368234815	Induction of severe inflammation.
	*IFN/IL-28*	rs12979860	Promotes inflammation and fibrosis (it is predictive for advanced stage of the disease).
	*PCSK7*	rs236918	Liver damage and altered fibrogenesis association.
	*MERTK*	rs4374383	Protection against fibrosis.
	*HSD17B13*	rs72613567	Reduced risk of NASH (but not steatosis).

Genetic variants identified as associated with MAFLD (Metabolic (dysfunction) associated fatty liver disease) and NASH (Non-alcoholic steatohepatitis) encodes for genes involved in several metabolism pathways. Some variants seem able to protect from MAFLD onset and progression to NASH; unfortunately, many others promote hepatic steatosis and its worsening in severe inflammation and fibrosis. *PNPLA3, Patatin-like phospholipase domain-containing 3; MBOAT 7, Membrane-bound O-acyltransferase domain-containing 7; TM6SF2, Transmembrane 6 superfamily member 2; GCKR, glucokinase regulator; PPP1R3B, protein phosphatase 1 regulatory subunit 3B; APOB, APOB100; SOD2, manganese-dependent superoxide dismutase; UCP2, Uncoupling protein 2; UCP3, Uncoupling protein 3; MARC1, Mitochondrial Amidoxime Reducing Component 1; TLR4, toll-like receptor 4; IFNL4, Interferon Lambda 4; PCSK7, Proprotein convertase subtilisin/Kexin type 7; MERTK, Mer T kinase;* HSCs, Hepatic stellate cells; HCC, Hepatocellular carcinoma; VLDL, Very-low-density lipoproteins.

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
