# Peer review of "Nutrigenomics and Nutrigenetics in Metabolic- (Dysfunction) Associated Fatty Liver Disease: Novel Insights and Future Perspectives"

_nutrients, 2021, doi:10.3390/nu13051679_

Round 1

Reviewer 1 Report

Review comments to the author:

In the reviewed manuscript, the author presents a comprehensive overview describing what is the role of nutrients, or be more specific nutrigenomic and nutrigenetic, play in metabolic associated fatty liver disease. The connection between the nutrigenetic and nutrigenomic in MAFLD to immunity is very novel and interesting. Overall, the structure is well-organized, and the flow is easy to follow. This work has the potential to contribute to the field, but the future perspective part needs to be further addressed.

Major concern:

From the future perspective part, it seems like the author trying to bring up the importance of immunity as part of the tangled network and highlight a future study direction toward the connection between nutrients and immunity. But the evidence and example are limited. In this part, the author mainly and only focuses on oxLDL and its impact on macrophages. However, the immunity has its complicated network and includes many different cell types. Moreover, macrophage only account for very small portion of resident immune repertoire. Thus, I strongly suggest the author to give more evidence and example besides macrophage to strengthen their standpoint.

Overall, the work is very impressive.

Author Response

We thank the reviewer very much for the careful revision of our manuscript. In this submitted version, we took advantage of the corrections received and modified the text accordingly.

Actually we mainly focused the attention on the interconnection between nutrients, gene and Trained Immunity that currently represents an absolute novelty in this field, never explored before. For this reason, the text appears for the greater part referred to macrophages pathobiology that until now, following the current knowledge, represents the main cellular subtype involved in this processes, basing on the oxLDL stimulation (that represents the most interesting trigger linking TI activity and metabolic disorders like MAFLD). However, we totally agree with the concern highlighted by the reviewer. Overall, the involvement of the immune system in this particular context is very deep and difficult to analyse in a correct, comprehensive and “brief” paragraph on “future perspective” considering, also, that it’s for the greater part already explored up to not be considered properly as novelty. However, we modified the paragraph inserting a focus on other mechanisms involving other cellular subtypes.

Reviewer 2 Report

The authors have done an interesting and urgent review of the literature trying to discuss important aspects related to the field of Nutrigenomics and Nutrigenetics in MAFLD and future perspectives. However, the reviewer have many important and relevant suggestions in order to improve the content and relevance of such scientific contribution, as pointed below:

  1. Nutrigenetics could be also be influenced by epigenetics (e.g. when certain SNPs effect/impact will depend on epigenetic markers to express the respective gene). E.g. Dayeh TA et al., 2013 (Diabetologia. 2013 May;56(5):1036-46. ). Please acknowledge this aspect in your Figure 2.
  2. Line 125: continue and include within the"scientific knowledge" statement if you are referring to filed of MAFLD or the whole filed which at some point could be applicable (potentially be applied) to MAFLD due to aspect you will try to discuss/explain through your review.  
  3. Could you include some discussion about any possibilities of using genetic score at the moment for MAFLD prediction/progression? And how far we are from this, is there any % of prediction/explanation for each variant at the moment? Any limitations for the use of this approach? This could be discussed between lines, e.g., 225-228.
  4. The authors should mention other epigenetics studies from the NASH/NAFLD field. Even if they are mentioned in reviews, the original articles should be presented (e.g. Kitamoto  et al., 2015 J Hepatol;  de Mello et al., 2017 Epigenetics; de Mello etal., 2021 Liver Int). This should be included in within your section 2.2.
  5. Within these several epigenetic phenomena that relates to the risk factors of the disease that are modifiable by diet (e.g. insulin resistance, serum colesterol) could be discussed as potential nutrigenomic approach. 
  6. Line 342: please acknowledge that these responses may eventually vary according to genotype at least related to glycemia, insulin secretion and inflammation (see Lankinen et al., 2019 and 2021 in teh case for FADS genotype), specially teh later as a risk factor for CVDs.
  7. Data on the effect of diet on risk factors or metabolic disturbances that accompany NAFLD (NASH and/or steatosis) should be at least summarized as important background for future studies in section 3. 
  8. Section 3 should be divided according to nutrient and have a conclusion paragraph in case of interaction. The same for section 4, in which Fig. 3 should be also amended a suggested below. 
  9. Figure 3. Please include "Proposed figure for" or similar statement in the legend of your figure. More explanation about dietary sources of mentioned fatty acids should be provided (= translated).
  10. Line 466: please replace "assumption" by "consumption".
  11. line 503-523: It's not clear from where the evidence comes when implying MBOAT7 interaction with dietary intake to prevent or increase NAFLD/NASH/steatosis.
  12. line 541-544: another place worth to mention gene score
  13. line 566-568: this information here needs to be translated to diet. Which kind of diets promote these "overloads".
  14. Before section 5 it would help if another section where the authors discuss immunity and diet/inflammation to make a more solid background for the aim of the current section 5. Or at least discuss immunity at the beginning of this section.

Author Response

We thank the reviewer very much for the careful revision of our manuscript. In this submitted version, we took advantage of the corrections received and modified the text accordingly:

  1. As properly suggested we modified the figure 2;
  2. We modified the text according to the reviewer suggestion;
  3. As suggested, we modified the last part of the paragraph 2.1 focusing the attention on the role of genetic score in prediction/progression of MAFLD;
  4. We totally agreed with the reviewer. As requested, we added some crucial information related to the cited articles in the main text.
  5. We thank the reviewer for the suggestion. We added this consideration on the nutrigenomic, epigenetic and clinical practice relationship at the end of the section 2.2 as a bridge to the next one.
  6. We added as requested in the specific section the important findings highlighted by FADSDIET1 and 2 trials in this context.
  7. We interpreted the request of the reviewer as oriented to highlight in a more incisive way the actual scientific knowledge regarding the possibility of a tailored gene-based dietary regimen to treat the risk factors surrounding MAFLD. We briefly mentioned the most significant evidence on this purpose even if the lack of sufficient scientific proves to support definitive statements on this topic remains still remarkable. Of course, as properly indicated and inserted in the main text, it represents an important background for future studies.
  8. We divided the paragraph 3 and 4 in some subsections according to the suggestion of the reviewer
  9. We modified the figure as requested
  10. We modified as suggested
  11. We properly corrected the mistake
  12. We took advantage from the suggestion implementing the manuscript with some knowledge deriving from the gene score applicability.
  13. Maybe the identified section results a bit confusing. The sentence: “In line with this, it could be interesting to focus the attention on the immunomodulatory effects of cholesterol, palmitate and FFAs overload” was not referred to a specific type of diet. In fact, it was only an introductory sentence for the next section in which we analyzed separately the role of cholesterol, palmitate and FFA overload in this context, basing on the scientific knowledge properly referred. In order to avoid any type of misunderstanding we eliminated the sentence.
  14. We agree with the reviewer and added a brief extra section at the beginning of paragraph 5 focused on diet and immunity interaction. During the work plan for this paper we decided, at the beginning, to avoid the opportunity to debate this relationship because considered out of the main scope of the review, rather focusing the attention on nutrigenomics and nutrigenetics influences on immune functioning and highlighting, particularly, the concept of trained immunity in this context, that represents an absolute novelty and, for this reason, need to be explored and mentioned in a “future perspectives” paragraph. However, thanks to the reviewer observation we retain now this new version clearer and more explanatory.

Round 2

Reviewer 2 Report

The authors did a very good job improving the manuscript while taking into consideration the comments from this reviewer. I would like through, to still point out for the fact that SCFAs are derived from intestinal microbial fermentation of indigestible foods, so that as yourselves explained a bit later in the text, that this leads to SCFAs production. Therefore, I suggest  the authors not to mention that SCFAs come from food as it is now in lines 692-3. 

Also, English language should be revised in order to concise the sentences, that still in various parts of the manuscript, are rather long. For example, between lines 734-739.

Author Response

Response to reviewer 2 (round 2)

We thank the reviewer very much for the careful revision of our manuscript. We are totally agree with the highlighted minor criticisms. We modified the sections as requested and shortened the long sentences to improve the readability.